# Fooling Detection Alone is Not Enough: Adversarial Attack against Multiple Object Tracking

**Yunhan Jia**[*1], **Yantao Lu**[*2], **Junjie Shen**[3], **Qi Alfred Chen**[3], **Hao Chen**[4],
**Zhenyu Zhong**[5], **Tao Wei**[6]
[1]Independent Researcher, [2]Syracuse University, [3]UC Irvine, [4]UC Davis,
[5]Baidu X-Lab, [6] Peking University
jack0082010@gmail.com,yl25@syr.edu,{alfchen,junjies1}@ucr.edu, chen@ucdavis.edu,
edwardzhong@baidu.com, lenx.wei@gmail.com,

## Abstract

Recent work in adversarial machine learning started to focus on the visual perception in autonomous driving and studied Adversarial Examples (AEs) for object detection models. However, in such visual perception pipeline the detected objects must also be tracked, in a process called Multiple Object Tracking (MOT), to build the moving trajectories of surrounding obstacles. Since MOT is designed to be robust against errors in object detection, it poses a general challenge to existing attack techniques that blindly target objection detection: we find that a success rate of over 98% is needed for them to actually affect the tracking results, a requirement that no existing attack technique can satisfy. In this paper, we are the first to study adversarial machine learning attacks against the complete visual perception pipeline in autonomous driving, and discover a novel attack technique, tracker hijacking, that can effectively fool MOT using AEs on object detection. Using our technique, successful AEs on as few as one single frame can move an existing object in to or out of the headway of an autonomous vehicle to cause potential safety hazards. We perform evaluation using the Berkeley Deep Drive dataset and find that on average when 3 frames are attacked, our attack can have a nearly 100% success rate while attacks that blindly target object detection only have up to 25%.

## 1    Introduction

Since the first Adversarial Example (AE) against traffic sign image classification discovered by Eykholt *et al.* (Eykholt et al., 2018), several research work in adversarial machine learning (Eykholt et al., 2017; Xie et al., 2017; Lu et al., 2017a;b; Zhao et al., 2018b; Chen et al., 2018; Cao et al., 2019) started to focus on the context of visual perception in autonomous driving, and studied AEs on object detection models. For example, Eykholt *et al.* (Eykholt et al., 2017) and Zhong *et al.* (Zhong et al., 2018) studied AEs in the form of adversarial stickers on stop signs or the back of front cars against YOLO object detectors (Redmon & Farhadi, 2017), and performed indoor experiments to demonstrate the attack feasibility in the real world. Building upon these work, most recently Zhao *et al.* (Zhao et al., 2018b) leveraged image transformation techniques to improve the robustness of such adversarial sticker attacks in outdoor settings, and were able to achieve a 72% attack success rate with a car running at a constant speed of 30 km/h on real roads.

While these results from prior work are alarming, object detection is in fact only the first half of the visual perception pipeline in autonomous driving, or in robotic systems in general — in the second half, the detected objects must also be tracked, in a process called *Multiple Object Tracking (MOT)*, to build the moving trajectories, called *trackers*, of surrounding obstacles. This is *required* for the subsequent driving decision making process, which needs the built trajectories to predict future moving trajectories for these obstacles and then plan a driving path accordingly to avoid collisions with them. To ensure high tracking accuracy and robustness against errors in object detection, in MOT only the detection results with sufficient consistency and stability across multiple frames can be included in the tracking results and actually influence the driving decisions. Thus, MOT in the visual

---
[*]Equal contribution

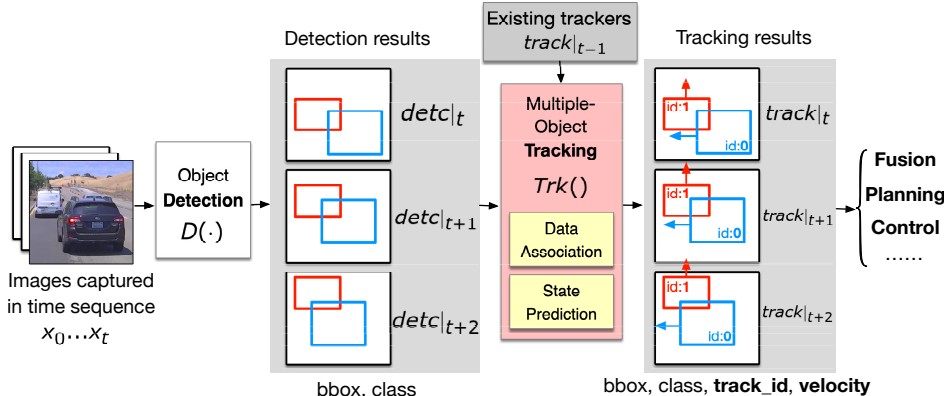

Figure 1: The complete visual perception pipeline in autonomous driving, i.e., both object detection and Multiple Object Tracking (MOT) (Baidu; Kato et al., 2018; 2015; Zhao et al., 2018a; Ess et al., 2010; MathWorks; Udacity).

perception of autonomous driving poses a general challenge to existing attack techniques that blindly target objection detection. For example, as shown by our analysis later in §4, an attack on objection detection needs to succeed consecutively for at least 60 frames to fool a representative MOT process, which requires an at least $98\%$ attack success rate (§4). To the best of our knowledge, no existing attacks on objection detection can achieve such a high success rate (Eykholt et al., 2017; Xie et al., 2017; Lu et al., 2017a;b; Zhao et al., 2018b; Chen et al., 2018).

In this paper, we are the first to study adversarial machine learning attacks considering the *complete* visual perception pipeline in autonomous driving, i.e., both object detection and object tracking, and discover a novel attack technique, called *tracker hijacking*, that can effectively fool the MOT process using AEs on object detection. Our key insight is that although it is highly difficult to directly create a tracker for fake objects or delete a tracker for existing objects, we can carefully design AEs to attack the tracking error reduction process in MOT to deviate the tracking results of existing objects towards an attacker-desired moving direction. Such process is designed for increasing the robustness and accuracy of the tracking results, but ironically, we find that it can be exploited by attackers to substantially alter the tracking results. Leveraging such attack technique, successful AEs on as few as *one single frame* is enough to move an existing object in to or out of the headway of an autonomous vehicle and thus may cause potential safety hazards.

We select 20 out of 100 randomly sampled video clips from the Berkeley Deep Drive dataset for evaluation. Under recommended MOT configurations in practice (Zhu et al., 2018) and normal measurement noise levels, we find that our attack can succeed with successful AEs on as few as *one frame*, and 2 to 3 consecutive frames on average. We reproduce and compare with previous attacks that blindly target object detection, and find that when attacking 3 consecutive frames, our attack has a nearly 100% success rate while attacks that blindly target object detection only have up to 25%.

**Contributions.** In summary, this paper makes the following contributions:

- We are the first to study adversarial machine learning attacks considering the complete visual perception pipeline in autonomous driving, i.e., both object detection and MOT. We find that without considering MOT, an attack blindly targeting object detection needs at least a success rate of 98% to actually affect the complete visual perception pipeline in autonomous driving, which is a requirement that no existing attack technique can satisfy.

- We discover a novel attack technique, tracker hijacking, that can effectively fool MOT using AEs on object detection. This technique exploits the tracking error reduction process in MOT, and can enable successful AEs on as few as one single frame to move an existing object in to or out of the headway of an autonomous vehicle to cause potential safety hazards.

- The attack evaluation using the Berkeley Deep Drive dataset shows that our attack can succeed with successful AEs on as few as one frame, and only 2 to 3 consecutive frames on average, and when 3 consecutive frames are attacked, our attack has a nearly 100% success rate while attacks that blindly target object detection only have up to 25%.

- Code and evaluation data are all available at GitHub (Github).

## 2 BACKGROUND AND RELATED WORK

**Adversarial examples for object detection.** Since the first physical adversarial examples against traffic sign classifier demonstrated by Eykholt *et al.* (Eykholt et al., 2018), several work in adversarial machine learning (Eykholt et al., 2017; Xie et al., 2017; Lu et al., 2017a;b; Zhao et al., 2018b; Chen et al., 2018) have been focused on the visual perception task in autonomous driving, and more specifically, the object detection models. To achieve high attack effectiveness in practice, the key challenge is how to design robust attacks that can survive distortions in real-world driving scenarios such as different viewing angles, distances, lighting conditions, and camera limitations. For example, Lu *et al.* (Lu et al., 2017a) shows that AEs against Faster-RCNN (Ren et al., 2015) generalize well across a sequence of images in digital space, but fail in most of the sequence in physical world; Eykholt *et al.* (Eykholt et al., 2017) generates adversarial stickers that, when attached to stop sign, can fool YOLOv2 (Redmon & Farhadi, 2017) object detector, while it is only demonstrated in indoor experiment within short distance; Chen *et al.* (Chen et al., 2018) generates AEs based on expectation over transformation techniques, while their evaluation shows that the AEs are not robust to multiple angles, probably due to not considering perspective transformations (Zhao et al., 2018b). It was not until recently that physical adversarial attacks against object detectors achieve a decent success rate (70%) in fixed-speed (6 km/h and 30 km/h) road test (Zhao et al., 2018b).

While the current progress in attacking object detection is indeed impressive, in this paper we argue that in the actual visual perception pipeline of autonomous driving, object tracking, or more specifically MOT, is a integral step, and without considering it, existing adversarial attacks against object detection still cannot affect the visual perception results even with high attack success rate. As shown in our evaluation in §4, with a common setup of MOT, an attack on object detection needs to reliably fool at least 60 consecutive frames to erase one object (e.g., stop sign) from the tracking results, in which case even a 98% attack success rate on object detectors is not enough (§4).

**MOT background.** MOT aims to identify objects and their trajectories in video frame sequence. With the recent advances in object detection, *tracking-by-detection* (Luo et al., 2014) has become the dominant MOT paradigm, where the detection step identifies the objects in the images and the tracking step links the objects to the trajectories (*i.e.*, trackers). Such paradigm is widely adopted in autonomous driving systems today (Baidu; Kato et al., 2018; 2015; Zhao et al., 2018a; Ess et al., 2010; MathWorks; Udacity), and a more detailed illustration is in Fig. 1. As shown, each detected objects at time $t$ will be associated with a dynamic state model (e.g., position, velocity), which represents the past trajectory of the object ($track|_{t-1}$). A per-track Kalman filter (Baidu; Kato et al., 2018; Feng et al., 2019; Murray, 2017; Yoon et al., 2016) is used to maintain the state model, which operates in a recursive *predict-update* loop: the predict step estimates current object state according to a motion model, and the update step takes the detection results $detc|_t$ as *measurement* to update its state estimation result $track|_t$.

The association between detected objects with existing trackers is formulated as a bipartite matching problem (Sharma et al., 2018; Feng et al., 2019; Murray, 2017) based on the pairwise similarity costs between the trackers and detected objects, and the most commonly used similarity metric is the spatial-based cost, which measures the overlapping between bounding boxes, or bboxes (Baidu; Long et al., 2018; Xiang et al., 2015; Sharma et al., 2018; Feng et al., 2019; Murray, 2017; Zhu et al., 2018; Yoon et al., 2016; Bergmann et al., 2019; Bewley et al., 2016). To reduce errors in this association, an accurate velocity estimation is necessary in the Kalman filter prediction (Choi, 2015; Yilmaz et al., 2006). Due to the discreteness of camera frames, Kalman filter uses the velocity model to estimate the location of the tracked object in the next frame in order to compensate the object motion between frames. However, as described later in §3, such error reduction process unexpectedly makes it possible to perform tracker hijacking.

MOT manages tracker creation and deletion with two thresholds. Specifically a new tracker will be created only when the object has been constantly detected for a certain number of frames, this threshold will be referred to as the *hit count*, or $H$ in the rest of the paper. This helps to filter out occasional false positives produced by object detectors. On the other hand, a tracker will be deleted if no objects is associated with for a duration of $R$ frames, or called a *reserved age*. It prevents the tracks from being accidentally deleted due to infrequent false negatives of object detectors. The configuration of $R$ and $H$ usually depends on both the accuracy of detection models, and the frame rate (fps). Previous work suggest a configuration of $R = 2\cdot$ fps, and $H = 0.2\cdot$ fps (Zhu et al., 2018), which gives a $R = 60$ frames and $H = 6$ frames for a common 30 fps visual perception system. We

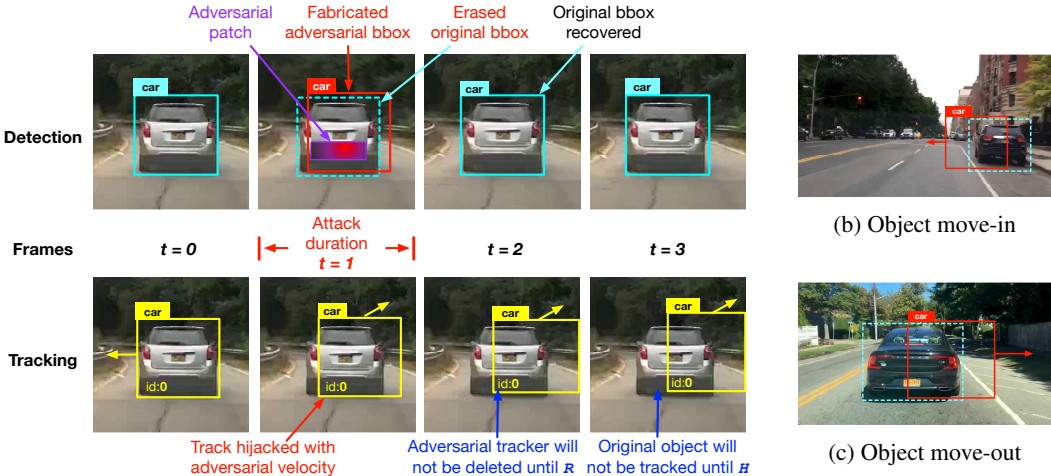

(a) Tracker hijacking attack overview

Figure 2: Description of the tracker hijacking attack flow (**a**), and two different attack scenarios: object move-in (**b**) and move-out (**c**), where tracker hijacking may lead to severe safety consequences including emergency stop and rear-end crashes.

will show in §4 that an attack that blindly targeting object detection needs to constantly fool at least 60 frames ($R$) to erase an object, while our proposed tracker hijacking attack can fabricate object that last for $R$ frames and vanish target object for $H$ frames in the tracking result by attacking as few as one frame, and only 2~3 frames on average ($S4$).

# 3 TRACKER HIJACKING ATTACK

**Scope.** This work focuses on the *track-by-detection* pipeline as described above, which has been recognized as the dominant MOT paradigm in recent literature (Long et al., 2018; Murray, 2017; Sharma et al., 2018; Luo et al., 2014) and MOT challenges (Dendorfer et al., 2019). A MOT approach can choose to include one or more similarity measures to match objects across frames. Common measures include bounding box overlaps, object appearances, visual representations, and other statistical measures (Luo et al., 2014). As the first study on the adversarial threats against MOT, we choose the *IoU-based Hungarian matching* (Sharma et al., 2018; Feng et al., 2019; Murray, 2017) as our target algorithm, as it is the most widely adopted and standardized similarity metric by not only very recent work (Long et al., 2018; Xiang et al., 2015; Feng et al., 2019), but also two real-world autonomous driving systems, i.e., Baidu Apollo (Baidu) and Autoware (Kato et al., 2018). This thus ensures the representativeness and practical significance of our work.

**Overview.** Fig. 2a illustrates the tracker hijacking attack discovered in this paper, in which an AE for object detection (e.g., in the form of adversarial patches on the front car) that can fool the detection result for as few as one frame can largely deviate the tracker of a target object (e.g., a front car) in MOT. As shown, the target car is originally tracked with a predicted velocity to the left at $t_0$. The attack starts at time $t_1$ by applying an adversarial patch onto the back of the car. The patch is carefully generated to fool the object detector with two adversarial goals: (1) erase the bounding box of target object from detection result, and (2) fabricate a bounding box with similar shape that is shifted a little bit towards an attacker-specified direction. The fabricated bounding box (red one in detection result at $t_1$) will be associated with the original tracker of target object in the tracking result, which we call a *hijacking* of the tracker, and thus would give a fake velocity towards the attacker-desired direction to the tracker. The tracker hijacking shown in Fig. 2a lasts for only one frame, but its adversarial effects could last tens of frames, depending on the MOT parameter $R$ and $H$ (introduced in §2). For example, at time $t_2$ after the attack, all detection bounding boxes are back to normal, however, two adversarial effects persist: (1) the tracker that has been hijacked with attacker-induced velocity *will not be deleted until a reserved age ($R$) has passed*, and (2) the target object, though is recovered in the detection result, *will not be tracked until a hit count ($H$) has reached*, and before that the object remains missing in the tracking result. However, it's important to note that our attack may not always succeed with one frame in practice, as the recovered object may still be associated with its original tracker, if the tracker is not deviated far enough from the object's true position during a short attack

duration. Our empirical results show that our attack usually achieves a nearly 100% success rate when 3 consecutive frames are successfully attacked using AE (§4).

Such persistent adversarial effects may cause severe safety consequences in self-driving scenarios. We highlight two attack scenarios that can cause emergency stop or even a rear-end crashes:

**Attack scenario 1: Target object move-in.** Shown in Fig. 2b, an adversarial patch can be placed on roadside objects, *e.g.*, a parked vehicle to deceive visual perception of autonomous vehicles passing by. The adversarial patch is generated to cause a translation of the target bounding box towards the center of the road in the detection result, and the hijacked tracker will appear as a moving vehicle cutting in front in the perception of the victim vehicle. This tracker would last for 2 seconds if $R$ is configured as $2\cdot$ fps as suggested in (Zhu et al., 2018), and tracker hijacking in this scenario could cause an emergency stop and potentially a rear-end crash.

**Attack scenario 2: Target object move-out.** Similarly, tracker hijacking attack can also deviate objects in front of the victim autonomous vehicle away from the road to cause a crash as shown in Fig. 2c. Adversarial patch applied on the back of front car could deceive MOT of autonomous vehicle behind into believing that the object is moving out of its way, and the front car will be missing from the tracking result for a duration of $200ms$, if $H$ uses the recommended configuration of $0.2\cdot$ fps (Zhu et al., 2018). This may cause the victim autonomous vehicle to crash into the front car.

### 3.1 Attack Methodology

---
**Algorithm 1** Tracker Hijacking Attack

---
**Input:** Video image sequence $X = [x_0, x_1, ..., x_n]$; object detector $D(\cdot)$; MOT algorithm $Trk(\cdot)$;
**Input:** Index of target object to be hijacked $K$, attacker-desired directional velocity $\vec{v}$, adversarial patch area as a mask matrix $patch$.
**Output:** Sequence of adversarial examples $X' = [x'_1, ..., x'_r]$ required for a successful attack.
**Initialization** $X' \leftarrow \{\}$, $detc|_0 \leftarrow D(x_0)$, $track|_0 \leftarrow \{current\_tracks\}$

1: **for** $t = 1$ to $n$ **do**
2:    $detc|_t \leftarrow D(x_t)$
3:    **if** $detc|_t[K]$ matches $track|_{t-1}[K]$ **then**    ▷ target object matches with an existing tracker
4:        find position $pos$ to place fabricated bbox with Eq. 1

$$pos \leftarrow \text{FindPos}(Trk(\cdot), track|_{t-1}, K, \vec{v}, patch) \quad \text{see Alg. 2 in Appendix}$$

5:        generate adversarial frame $x'$ with Eq. 3    ▷ attack object detector with specialized loss

$$x'_t \leftarrow \text{GenerateAdv}(x, D(\cdot), pos, K, patch) \quad \text{see Alg. 3 in Appendix}$$

6:        $X' \xleftarrow{+} x'_t$
7:    **else**
8:        **return** $X'$    ▷ attack succeeds when target object is not associated with original tracker
9:    **end if**
10:    $track|_t \leftarrow Trk(track|_{t-1}, D(x'_t))$                ▷ update current tracker with adversarial frame
11: **end for**

---

**Targeted MOT design.** Our attack targets on first-order Kalman filter, which predicts a state vector containing position and velocity of detected objects over time. For the data association, we adopt the mostly widely used Intersection over Union (IoU) as the similarity metric, and the IoU between bounding boxes are calculated by Hungarian matching algorithm (Luetteke et al., 2012) to solve the bipartite matching problem that associates bounding boxes detected in consecutive frames with existing trackers. Such combination of algorithms in the MOT is the most common in previous work (Long et al., 2018; Xiang et al., 2015; Sharma et al., 2018) and real-world systems (Baidu).

We now describe our methodology of generating an adversarial patch that manipulates detection results to hijack a tracker. As detailed in Alg. 1, given a targeted video image sequence, the attack iteratively finds the minimum required frames to perturb for a successful track hijack, and generates the adversarial patches for these frames. In each attack iteration, an image frame in the original video clip is processed, and given the index of target objects $K$, the algorithm finds an optimal position to place the adversarial bounding box $pos$ in order to hijack the tracker of target object by solving Eq. 1. The attack then constructs adversarial frame against object detection model with an adversarial

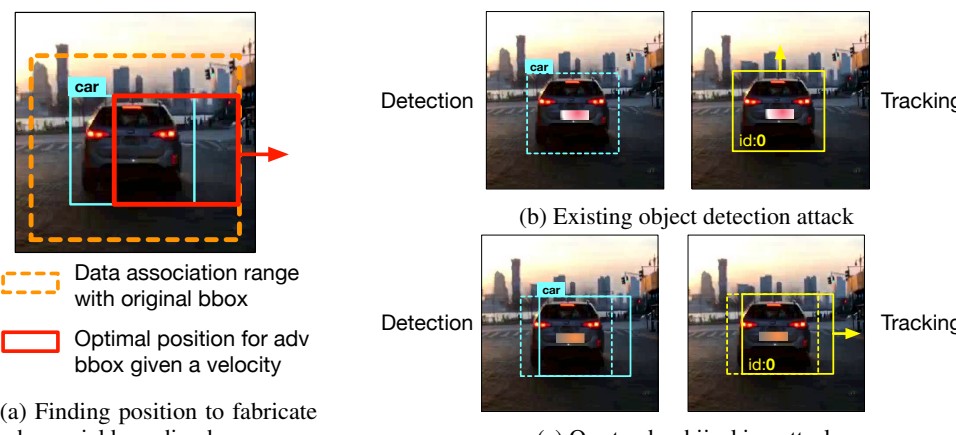

(a) Finding position to fabricate adversarial bounding box

(b) Existing object detection attack

(c) Our tracker hijacking attack

Figure 3: Comparison between previous object detection attack and our tracker hijacking attack. Previous attack that simply erase the bbox has no impact on the tracking output (**b**), while tracker hijacking attack that fabricates bbox with carefully chosen position successfully redirects the tracker towards attacker-specified direction (**c**).

patch, using Eq. 3 as the loss function to erase the original bounding box of target object and fabricate the adversarial bounding box at the given location. The tracker is then updated with the adversarial frame that deviates the tracker from its original direction. If the target object in the next frame is not associate with its original tracker by the MOT algorithm, attack has succeeded; otherwise, this process is repeated for the next frame. We discuss two critical steps in this algorithm below, and please refer to the Appendix A for the complete implementation of the algorithm.

**Finding optimal position for adversarial bounding box.** To deviate the tracker of a target object $K$, besides removing its original bounding box $detc|_t[K]$, the attack also needs to fabricate an adversarial box with a shift $\delta$ towards a specified direction. This turns into an optimization problem (Eq. 1) of finding the translation vector $\delta$ that maximizes the cost of Hungarian matching ($\mathcal{M}(\cdot)$) between the detection box and the existing tracker so that the bounding box is still associated with its original tracker ($\mathcal{M} \le \lambda$), but the shift is large enough to give an adversarial velocity to the tracker. Note that we also limit the shifted bounding box to be overlapped with the $patch$ to facilitate adversarial example generation , as it's often easier for adversarial perturbations to affect prediction results in its proximity, especially in physical settings (Chen et al., 2018).

$$\max_{\delta} \mathcal{M}(detc|_t[K] + \delta, track|_{t-1}[K])$$
$$s.t. \ \mathcal{M} \le \lambda, IoU(detc|_t[K] + \delta, patch) \ge \gamma \tag{1}$$

**Generating adversarial patch against object detection.** Similar to the existing adversarial attacks against object detection models (Chen et al., 2018; Eykholt et al., 2018; Zhao et al., 2018b), we also formulate the adversarial patch generation as an optimization problem shown in Eq. 3 in Appendix. Existing attacks without considering MOT directly minimize the probability of target class (e.g., a stop sign) to erase the object from detection result. However, as shown in Fig. 3b, such AEs are highly ineffective in fooling MOT as the tracker will still track for $R$ frames even after the detection bounding box is erased. Instead, the loss function of our tracker hijacking attack incorporates two optimization objectives: (1) minimizes the target class probability to erase the bounding box of target object; (2) fabricates the adversarial bounding box at the attacker-desired location and in the specific shape to hijack the tracker. Details of our algorithm can be found in Appendix A, and the implementation can be found at (Github).

## 4 ATTACK EVALUATION

In this section, we describe our experiment settings for evaluating the effectiveness of our tracker hijacking attack, and comparing it with previous attacks that blindly attack object detection in detail.

### 4.1 EXPERIMENT METHODOLOGY

**Evaluation metrics.** We define a successful attack as that *the detected bounding box of target object can no longer be associated with any of the existing trackers when attack has stopped*. We measure the effectiveness of our track hijacking attack using the minimum number of frames that the AEs on

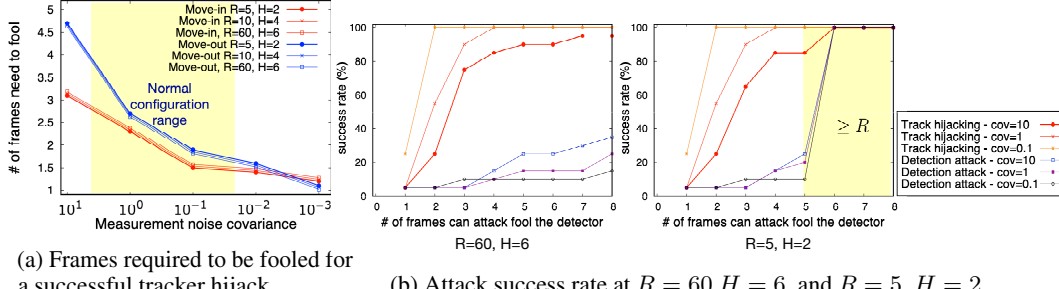

(a) Frames required to be fooled for a successful tracker hijack

(b) Attack success rate at $R = 60$ $H = 6$, and $R = 5$, $H = 2$

Figure 4: In normal measurement noise covariance range (**a**), our tracker hijacking attack would require the AE (adversarial example) to fool only 2~3 consecutive frames on average to successfully deviate the target tracker despite the $(R, H)$ settings. We also compare the success rate of tracker hijacking with previous adversarial attack against object detectors only under different attacker capabilities, *i.e.*, the number of consecutive frames the AE can reliably fool the object detector (**b**). Tracker hijacking achieves superior attack success rate (100%) even by fooling as few as 3 frames, while previous attack is only effective when the AE can reliably fools at least $R$ consecutive frames.

object detection need to succeed. The attack effectiveness highly depends on the difference between the direction vector of the original tracker and adversary's objective. For example, attacker can cause a large shift on tracker with only one frame if choosing the adversarial direction to be opposite to its original direction, while it would be much harder to deviate the tracker from its established track, if the adversarial direction happens to be the same as the target's original direction. To control the variable, we measure the number of frames required for our attack in two previous defined attack scenarios: target object move-in and move-out. Specifically, in all move-in scenarios, we choose the vehicle parked along the road as target, and the attack objective is to move the tracker to the center, while in all move-out scenarios, we choose vehicles that are moving forward, and the attack objective is to move the target tracker off the road.

**Dataset selection.** We randomly sampled 100 video clips from Berkeley Deep Drive dataset (Yu et al., 2018), and then manually selected 10 suitable for the object move-in scenario, and another 10 for the object move-out scenario. For each clip, we manually label a target vehicle and annotate the patch region to be a small area at its back as shown in Fig. 3c. All videos are 30 frames per second.

**Implementation details.** We implement our targeted visual perception pipeline using Python, with YOLOv3 (Redmon & Farhadi, 2018) as the object detection model due to its high popularity among in real-time systems. For the MOT implementation, we use the Hungarian matching implementation called `linear_assignment` in the `sklearn` package for the data association, and we provide a reference implementation of Kalman filter based on the one used in OpenCV (OpenCV).

The effectiveness of attack depends on a configuration parameter of Kalman filter, called *measurement noise covariance* (*cov*). *cov* is an estimation about how much noise is in the system, a low *cov* value would give Kalman filter more confidence on the detection result at time $t$ when updating the tracker, while a high *cov* value would make Kalman filter to place trust more on its own previous prediction at time $t - 1$ than that at time $t$. We give a detailed introduction of configurable parameters in Kalman filter in §2 of our Appendix B. This measurement noise covariance is often tuned based on the performance of detection models in practice. We evaluate our approach under different *cov* configurations ranging from very small ($10^{-3}$) to very large (10) as shown in Fig. 4a, while *cov* is usually set between 0.01 and 10 in practice (Baidu; Kato et al., 2018).

## 4.2 EVALUATION RESULTS

**Effectiveness of tracker hijacking attack.** Fig. 4a shows the average number of frames that the AEs on object detection need to fool for a successful track hijacking over the 20 video clips. Although a configuration with $R = 60$ and $H = 6$ is recommended when fps is 30 (Zhu et al., 2018), we still test different reserved age ($R$) and hit count ($H$) combinations as real-world deployment are usually more conservative and use smaller $R$ and $H$ (Baidu; Kato et al., 2018). The results show that tracker hijacking attack only requires successful AEs on object detection in 2 to 3 consecutive frames on average to succeed despite the $(R, H)$ configurations. We also find that even with a successful AE on only one frame, our attack still has 50% and 30% success rates when *cov* is 0.1 and 0.01 respectively.

Interestingly, we find that object move-in generally requires less frames compared with object move-out. The reason is that the parked vehicles in move-in scenarios (Fig. 2b) naturally have a moving-away velocity relative to the autonomous vehicle. Thus, compared to move-out attack, move-in attack triggers a larger difference between the attacker-desired velocity and the original velocity. This makes the original object, once recovered, harder to associate correctly, making hijacking easier.

**Comparison with attacks that blindly target object detection.** Fig. 4b shows the success rate of our attack and previous attacks that blindly target object detection (denoted as *detection attack*). We reproduced the recent adversarial patch attack on object detection from Zhong *et al.* (Zhong et al., 2018), which targets the autonomous driving context and showed effectiveness using real-world car testing. In this attack, the objective is to erase the target class from the detection result of each frame. Evaluated under two $(R, H)$ settings, we find that our tracker hijacking attack achieves superior attack success rate (100%) even by attacking as few as 3 frames, while the detection attack needs to reliably fool at least $R$ consecutive frames. When $R$ is set to 60 according to the frame rate of 30 fps, the detection attack needs to have an adversarial patch that can constantly succeed at least 60 frames while the victim autonomous vehicle is driving. This means an over $98.3\%$ ($\frac{59}{60}$) AE success rate, which has never been achieved or even got close to in prior work (Zhao et al., 2018b; Eykholt et al., 2017; Chen et al., 2018; Lu et al., 2017a). Note that the detection attack still can have up to ~25% success rate before $R$. This is because the detection attack causes the object to disappear for some frames, and when the vehicle heading changes during such disappearing period, it is still possible to cause the original object, when recovered, to misalign with the tracker predication in the original tracker. However, since our attack is designed to intentionally mislead the tracker predication in MOT, our success rate is substantially higher (3-4×) and can reach 100% with as few as 3 frames attacked.

## 5 DISCUSSION AND FUTURE WORK

**Implications for future research in this area.** Today, adversarial machine learning research targeting the visual perception in autonomous driving, no matter on attack or defense, uses the accuracy of objection detection as the *de facto* evaluation metric (Luo et al., 2014). However, as concretely shown in our work, without considering MOT, successful attacks on the detection results alone do not have direct implication on equally or even closely successful attacks on the MOT results, the final output of the visual perception task in real-world autonomous driving (Baidu; Kato et al., 2018). Thus, we argue that future research in this area should consider: (1) using the MOT accuracy as the evaluation metric, and (2) instead of solely focusing on object detection, also studying weaknesses specific to MOT or interactions between MOT and object detection, which is a highly under-explored research space today. This paper marks the first research effort towards both directions.

**Practicality improvement.** Our evaluation currently are all conducted digitally with captured video frames, while our method should still be effective when applied to generate physical patches. For example, our proposed adversarial patch generation method can be naturally combined with different techniques proposed by previous work to enhance reliability of AEs in the physical world (*e.g.*, non-printable loss (Sharif et al., 2016) and expectation-over-transformation (Athalye et al., 2017)). We leave this as future work.

**Generality improvement.** Though in this work we focused on MOT algorithm that uses IoU based data association, our approach of finding location to place adversarial bounding box is generally applicable to other association mechanisms (e.g., appearance-based matching). Our AE generation algorithm against YOLOv3 should also be applicable to other object detection models with modest adaptations. We plan to provide reference implementations of more real-world end-to-end visual perception pipelines to pave the way for future adversarial learning research in self-driving scenarios.

## 6 CONCLUSION

In this work, we are the first to study adversarial machine learning attacks against the complete visual perception pipeline in autonomous driving, i.e., both object detection and MOT. We discover a novel attack technique, tracker hijacking, that exploits the tracking error reduction process in MOT and can enable successful AEs on as few as one frame to move an existing object in to or out of the headway of an autonomous vehicle to cause potential safety hazards. The evaluation results show that on average when 3 frames are attacked, our attack can have a nearly 100% success rate while attacks that blindly target object detection only have up to 25%. The source code and data is all available at (Github).

Our discovery and results strongly suggest that MOT should be systematically considered and incorporated into future adversarial machine learning research targeting the visual perception in autonomous driving. Our work initiates the first research effort along this direction, and we hope that it can inspire more future research into this largely overlooked research perspective.

## ACKNOWLEDGMENTS

We would like to thank the anonymous reviewers for providing valuable feedback on our work. This research was supported in part by the National Science Foundation under grants CNS-1850533 and CNS-1932464.

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

## A   TRACK HIJACKING ATTACK DETAILS

Given the targeted video image sequence, track hijacking attack iteratively finds the minimum required frames to perturb for a successful hijack, and generates the adversarial patches for these frames. An image frame in the original video clip is given at each iteration and we use Alg. 2 to find an optimal position to place the adversarial bounding box $pos$ in order to hijack the tracker of target object.

The FINDPOS takes the existing tracking result $track|_{t-1}$, the detected objects $detc|_t$, the index of target object $K$, the attacker desired directional vector $\vec{v}$, the adversarial patch area $patch$ as input, and iteratively moves the bounding box along the direction of $\vec{v}$ while keeping some invariants: (1) the shifted bounding box shall still be associated with the original tracker of target object (Eq. 2); (2) the shifted bounding box shall always have overlap with the patch ($IoU(detc'[K], patch) > \gamma$). The `while` loop will end when the bounding box has been shifted to the farmost position from its original position along $\vec{v}$, where the invariants still hold . The intuition behind FINDPOS is that, in order for the tracker to loss track of the target object when attack has ended, attacker needs to deviate the bounding box of target object as far as possible within its original data association range.

$$\max_{\delta} \mathcal{M}(detc|_t[K] + \delta, track|_{t-1}[K])$$
$$s.t.\ \mathcal{M} \leq \lambda, IoU(detc|_t[K] + \delta, patch) \geqslant \gamma \tag{2}$$

---

**Algorithm 2** Track Hijacking Attack - Find fabricated bbox position

---

**Input:** Existing trackers $track|_{t-1}$; detection objects $detc|_t$; MOT algorithm $Trk(\cdot)$
**Input:** Index of target object to be hijacked $K$, attacker desired directional vector $\vec{v}$; adversarial patch area as a mask matrix $patch$
**Output:** fabricate bounding box position $pos$

  1: **procedure** FINDPOS
  2:     $detc' \leftarrow detc|_t$
  3:     $track' \leftarrow track|_{t-1}$
  4:     $k \leftarrow 1$
  5:     **while** $detc'[K]$ matches $track'[K]$ and $IoU(detc'[K], patch) > \gamma$ **do**
  6:        $detc'[K] \leftarrow track'[K] + v \cdot k$
  7:        $track' \leftarrow Tr(track', detc')$
  8:        $k = k + 1$
  9:     **end while**
10:     $pos = track'[K] + \vec{v} \cdot (k - 1)$
11:     **return** $pos$
12: **end procedure**

---

After the target bounding box location is identified, the next step is to generate adversarial patch against the object detection model. Similar to the existing adversarial attacks against object detection models (Chen et al., 2018; Eykholt et al., 2018; Zhao et al., 2018b), we also formulate the adversarial patch generation as an optimization problem shown in Eq. 3. Existing attacks without considering MOT directly minimize the probability of target class (e.g., a stop sign) to erase the target from detection result. However, as shown in Fig. 3b, such AEs are highly ineffective in fooling MOT as the tracker will still track for $R$ frames even after the detection bounding box is erased. Instead, the loss function of our tracker hijacking attack incorporates two loss terms: $\mathcal{L}_1$ minimizes the target class probability at given location to erase the target bounding box, where $\sum_{i=0}^{B} \mathbb{1}_i^{obj}$ identifies all bounding boxes ($B$) before non-max suppression (Neubeck & Van Gool, 2006), who contain the center location ($cx_t$, $cy_t$) of $pos$, while $C_i$ is the confidence score of bounding boxes; $\mathcal{L}_2$ controls the fabrication of adversarial bounding box at given center location ($cx_t$, $cy_t$) with given shape ($w_t$, $h_t$) to hijack the tracker. In the implementation, we use Adam optimizer to minimize the loss by iteratively perturbing the pixels along the gradient directions within the patch area, and the generation process stops when an adversarial patch that satisfies the requirements is generated. Note that the fabrication loss $\mathcal{L}_2$ needs only to be used when generating the first adversarial frame in a sequence to give the tracker an attacker-desired velocity $\vec{v}$, and then $\lambda$ can be set to 0 to only focus on erasing

target bounding box similar to previous work. Thus, our attack wouldn't add much difficulty to the optimization. The code of our implementation can be found at (Github).

$$\min_{\Delta \in patch} \mathcal{L}_1(x_t + \Delta) + \lambda \cdot \mathcal{L}_2(x_t + \Delta)$$

$$\mathcal{L}_1 = \sum_{i=0}^{B} \mathbb{1}_i^{obj} \cdot [C_i^2 - CrossEntropy(p_i, class_t)]$$

$$\mathcal{L}_2 = \sum_{i=0}^{B} \mathbb{1}_i^{obj} \cdot \{[(cx_i - cx_t)^2 + (cy_i - cy_t)^2] + [(\sqrt{w_i} - \sqrt{w_t})^2 + (\sqrt{h_i} - \sqrt{h_t})^2]$$

$$+ (1 - C_i)^2 + CrossEntropy(p_i, class_t)\}$$

(3)

Alg. 3 takes the adversarial bounding box position $pos$ for fabrication, and the original bounding box for vanish to generate an adversarial frame $x'$ whose perturbation is limited in the $patch$ area. Similar to the existing adversarial attacks against object detection models (Chen et al., 2018; Eykholt et al., 2018; Zhao et al., 2018b), we also formulate the adversarial patch generation as an optimization problem. First, the algorithm identifies all bounding boxes $i \in B$ in the intermediate result of object detection model before non-max suppression (Neubeck & Van Gool, 2006), and for all of them who contain the central point $c_x, c_y$ of $pos$ in its bounding box area, initialize $\mathbb{1}_i \leftarrow 1$, otherwise, $\mathbb{1}_i \leftarrow 0$. The algorithm then use Adam optimizer to minimize the loss $\mathcal{L}_1 + \lambda \mathcal{L}_2$ where $\mathcal{L}_1$ minimizes the target class probability in the vanish area, and $\mathcal{L}_2$ controls the fabrication of adversarial bounding box at given center location $(cx_t, cy_t)$ with given shape $(w_t, h_t)$ to hijack the tracker. Note that the fabrication loss $\mathcal{L}_2$ needs only to be used when generating the first adversarial frame in a sequence to give the tracker an attacker-desired velocity, and then $\lambda$ can be set to 0 to only focus on erasing target bounding box similar to previous work. Also note that when calculating the pixel gradient, we apply a mask $patch$ to the input $x$ to restrict the perturbation area. The attack stops when the maximum attack iteration has reached, and the adversarial example with the patch applied is returned. The implementation is available at (Github).

## B  KALMAN FILTER IMPLEMENTATION

The main idea behind Kalman filter is that the measurement result is not always reliable, and by combining a statistic noise model, the estimation can be more accurate than base on single measurement alone. This makes Kalman filter a natural fit for the *track-by-detection* pipeline, as MOT is intended to tolerate and correct the occasional errors in the detection result. The main principle of Kalman filter is represented as Eq. 4.

$$\hat{x}_k = K_k \cdot Z_k + (1 - K_k) \cdot \hat{x}_{k-1} \tag{4}$$

where $\hat{x}_k$ is the current state estimation, $K_k$ is the Kalman gain, $Z_k$ is the measurement value at state $k$, and $\hat{x}_{k-1}$ is the previous estimation. The equation shows that Kalman filter performs the state estimation using both the current measurement result and the previous estimation, while the Kalman gain $K_k$ is also a variable that will be updated by measurements.

In the MOT applications, the state estimations are the trackers, while the measurements are the detected bounding boxes at each frame. In this paper, we use first-order Kalman filter to track the central point location$(r, c)$ of bounding boxes, and first-order low-pass filter to track the width and length of bounding boxes with a decay factor 0.5, which is the same as Baidu Apollo self-driving platform's implementation (Baidu).

The update of the tracker states are updated with two steps: the time update, and the measurement update. The time update is performed as:

$$\hat{x}_k = \mathbf{F_k} \cdot \hat{x}_{k-1}$$
$$\mathbf{P_k} = \mathbf{F_k} \cdot \mathbf{P_{k-1}} \cdot \mathbf{F_k^T} + \mathbf{Q_k}$$

where $F_k$ is the first-order state transition model, and $\mathbf{P_k}$ is the posteriori error covariance matrix, which is a measure of the estimated accuracy of the state estimate. The $\mathbf{Q_K}$ is the covariance of the

---

**Algorithm 3** Track Hijacking Attack - Generate AE against object detection model

---

**Input:** Input image $x$; object detector $D(\cdot)$; all bounding boxes $B$ in $D(x)$ before non-max suppression; fabricated bbox position $pos$, attack iterations $N$;
**Input:** Index of target object to be hijacked $K$; adversarial patch area as a mask matrix $patch$.
**Output:** Adversarial example image $x'$.

1: **procedure** GENERATEADV
2:     $(c_x, c_y) \leftarrow$ central point of $pos$
3:     **for** all bboxes $i$ in $B$ **do**
4:         **if** bbox contains $(c_x, c_y)$ **then**
5:             $\mathbb{1}_i \leftarrow 1$
6:         **end if**
7:         $\mathbb{1}_i \leftarrow 0$
8:     **end for**
9:     $x' \leftarrow x$
10:     **for** $n = 0$ to $N$ **do**
11:         Calculate vanish loss $L_1$:

$$\mathcal{L}_1 = \sum_{i=0}^{B} \mathbb{1}_i^{obj} \cdot [C_i^2 - CrossEntropy(p_i, class_t)]$$

12:         Calculate fabricate loss $L_2$:

$$\mathcal{L}_2 = \sum_{i=0}^{B} \mathbb{1}_i^{obj} \cdot \{[(cx_i - cx_t)^2 + (cy_i - cy_t)^2] + [(\sqrt{w_i} - \sqrt{w_t})^2 + (\sqrt{h_i} - \sqrt{h_t})^2]$$
$$+ (1 - C_i)^2 + CrossEntropy(p_i, class_t)\}$$

13:         **if** $x$ is **not** the first frame to attack **then**
14:             $\lambda \leftarrow 0$
15:         **end if**
16:         Implement Adam optimizer to calculate pixel gradients

$$grad = Adam(patch \cdot x, \mathcal{L}_1 + \lambda\mathcal{L}_2)$$

17:         $x' \leftarrow x' + grad$
18:     **end for**
19:     **return** $x'$
20: **end procedure**

---

process noise. The measurement update is performed in the same loop as:

$$\mathbf{K_k} = \mathbf{P_k} \cdot \mathbf{H_k^T} \cdot (\mathbf{H_k} \cdot \mathbf{P_k} \cdot \mathbf{H_k^T} + \mathbf{R_k})^{-1}$$

$$\hat{x}_k' = \hat{x}_k + \mathbf{K}'(\vec{z}_k - \mathbf{H_k} \cdot \hat{x}_k)$$

$$\mathbf{P_k'} = \mathbf{P_k} - \mathbf{K}' \cdot \mathbf{H_k} \cdot \mathbf{P_k}$$

where $\mathbf{H_k}$ is the observation model, $\mathbf{R_k}$ the covariance of the observation model, and $\vec{z}_k$ the observation. In particular, denoting the coordinates of center point as $(r, c)$, we set the state vector $\vec{x}$ and state covariance matrix $\mathbf{P}$ as:

$$\vec{x} = \begin{bmatrix} p_r \\ v_r \\ p_c \\ v_c \end{bmatrix}, \mathbf{P} = \begin{bmatrix} \Sigma p_r p_r & \Sigma p_r v_r & \Sigma p_r p_c & \Sigma p_r v_c \\ \Sigma v_r p_r & \Sigma v_r v_r & \Sigma v_r p_c & \Sigma v_r v_c \\ \Sigma p_c p_r & \Sigma p_c v_r & \Sigma p_c p_c & \Sigma p_c v_c \\ \Sigma v_c p_r & \Sigma v_c v_r & \Sigma v_c p_c & \Sigma v_c v_c \end{bmatrix}$$

and we set the the state transition matrix $\mathbf{F}$ and the process covariance matrix $\mathbf{Q}$ as:

$$\mathbf{F} = \begin{bmatrix} 1 & dt & 0 & 0 \\ 0 & 1 & 0 & 0 \\ 0 & 0 & 1 & dt \\ 0 & 0 & 0 & 1 \end{bmatrix}, \mathbf{Q} = \begin{bmatrix} dt^4/4 & dt^3/2 & 0 & 0 \\ dt^3/2 & dt^2 & 0 & 0 \\ 0 & 0 & dt^4/4 & dt^3/2 \\ 0 & 0 & dt^3/2 & dt^2 \end{bmatrix}.$$

The observation matrix $\mathbf{H}$ and the measurement covariance $\mathbf{R}$ are set to be:

$$\mathbf{H} = \begin{bmatrix} 1 & 0 & 0 & 0 \\ 0 & 0 & 1 & 0 \end{bmatrix}$$

$$\mathbf{R} = cov \times \begin{bmatrix} 1 & 0 \\ 0 & 1 \end{bmatrix}$$

where the $cov$ is the variable we refered to as *measurement noise covariance* value we enumerated in our evaluation. From the expression of the Kalman gain in the measurement update process, we can see that the gain factor $K'$ is related to variations in $\mathbf{R}$. Identified by H.-G. Yeh *et al.* (Yeh, 1990), the Kalman gain can be regarded as a ratio of dynamic process to the measurement noise, *i.e.*, $K$ is proportional to $\frac{\mathbf{Q}}{cov \cdot \mathbf{I}}$. So when the $cov$ value is small, the object tracking response is relatively fast, and the tracking bounding boxes follow the detection boxes more closely; while when the $cov$ value is large, the Kalman filter trust more on its own estimation rather than the measurement, and the tracker is less responsive to the change of bounding boxes, which makes our track hijacking attack a little bit harder. In our paper, we empirically validated the impact of different $cov$ values **[0, 0.01, 0.1, 1, 10]** on the effectiveness of our attack, and found that under normal $cov$ configuration range (0.01 to 10), our attack can get a nearly 100% success rate by fooling 3 consecutive detection frames on average.

