# OpenReview forum: "Fooling Detection Alone is Not Enough: Adversarial Attack against Multiple Object Tracking"
_ICLR.cc/2020/Conference — Accept (Poster)_

### Official Review · AnonReviewer1 · 2019-10-23
**Official Blind Review #1**

**Rating:** 6

**Review:**

This paper is about conducting evasion attacks against Multiple Object Tracking (MOT) techniques. Compared to existing work on adversarial examples against object detection, to attack MOT techniques, the adversary needs to successfully fool multiple frames, and the authors show that by naively using existing attack approaches, the adversary needs to achieve 98% single-frame attack success rate to fool the tracking system, which is too hard for existing attack algorithms.  Therefore, this paper proposes a smart way of attacking MOT techniques by leveraging the properties of the tracking algorithm. In particular, they generate adversarial perturbations to remove the original bounding box while adding a fake bounding box that has some overlap with the original bounding box, so that the system will compute the movement of the object in a wrong way. They evaluate on videos in Berkeley Deep Drive dataset, and show that by attacking 2~3 frames, they can achieve nearly 100% attack success rate, while the attack success rate is only 25% if the tracking algorithm is not considered when crafting the attacks.

There has been a long line of work studying adversarial attacks against autonomous driving systems, which aims at revealing the security threat of such attacks in the real world. While the community has made promising progress, to my best knowledge, this paper is the first work considering attacking MOT techniques, and they suggest the importance of this attack scenario by demonstrating that attacking objection detection alone is not sufficient in terms of fooling the tracking technique. Meanwhile, the proposed attack method is interesting and effective, even if the perturbation is only bounded in a patch located at a reasonable place in the frame, e.g., the target car to attack.  Therefore, I lean towards accepting this paper, as it contributes to a new perspective of adversarial attacks.

However, I have some questions about evaluation.

1. To generate adversarial frames for the trajectory of the same car, is the adversarial patch inserted since the first frame and always stays the same, or different patches are needed to attack different frames of the same trajectory? I feel that if the adversarial patch needs to change a lot in the entire trajectory, then the practicality of the attack would be compromised.

2. How large does the adversarial patch need to be in order to successfully launch the attacks? And does the position of the patch affect the attack performance?

3. I wonder if the same adversarial patch may work beyond a single context, i.e., with the same patch, the same car can fool the tracking technique with different background scenes. I am not familiar with the details of the dataset in their evaluation, but it would be helpful to show relevant results if it is easy to simulate using their dataset.

------------
Post-rebuttal comments

I thank the authors for their response, and I keep my original assessment.
------------

**Experience Assessment:**

I have published in this field for several years.

**Review Assessment: Checking Correctness Of Derivations And Theory:**

I carefully checked the derivations and theory.

**Review Assessment: Checking Correctness Of Experiments:**

I carefully checked the experiments.

**Review Assessment: Thoroughness In Paper Reading:**

I read the paper thoroughly.

---

> ### Author Response · Authors · 2019-11-11
> **Authors' Response**
>
> Thank you for your insightful comments!
>
> - Is the inserted patch always stays the same?
> Currently in our evaluation, when comparing tracker hijacking attack and previous attacks, we generate a different patch for each video frame, which assumes an ideal situation for an attacker and is indeed different from realistic situation where the patch stays the same. However, since this assumption applies to both our attack and the detection attack, it does not affect our conclusion that our attack can significantly improve the attack effectiveness over previous attacks. In fact, in realistic situations where the patch stays the same in consecutive frames, the improvement of our attack over detection attack will be even larger than what is shown in figure 4(b), since to achieve the same attack success rate, the number of successfully-attacked consecutive frames required for detection attack is much larger than that for our attack.
>
> In addition, we would also like to note that there is a known technique called Expectations over Transformations (EoT) [1], which can help “average” the perturbations across different frames into one single perturbation that fools several frames reliably, and it is the basis for physical adversarial patch attacks in many previous work [1] [2] [3]. From the optimization perspective, the less frames to consider, the easier it is to generate a patch that fools all the frames using optimizer (e.g., Adam, SGD). Our tracker hijacking attack greatly reduced the number of frames required to be fooled compared to detection attacks from tens of them to only 2~3, and thus will make the generation of a universal patch much easier.
>
> - Does the position and size of the patch, and the background scene affect attack effectiveness? By how large?
> Yes, the size, position and background of the patch can affect the attack effectiveness in our case. Generally speaking, the larger the patch area, the easier it is to generate perturbations to fool the detection model. And theoretically when the patch is placed at the same position and under the same background scene in the attack generation process, the attack will be more effective. However, EoT techniques can also be used to make the patch much less sensitive to position and background changes. For example, a patch can be optimized using EoT to become effective under different distances and angles with respect to the camera [1]. Reliable physical patches have been demonstrated feasible by a long line of previous work focusing on image classifier and object detection models [1] [2] [3] [4].
>
> [1] https://arxiv.org/abs/1707.08945
> [2] https://www.usenix.org/conference/woot18/presentation/eykholt
> [3] https://arxiv.org/abs/1804.05810
> [4] https://arxiv.org/abs/1607.02533

---

### Official Review · AnonReviewer2 · 2019-10-23
**Official Blind Review #2**

**Rating:** 6

**Review:**

The paper addresses adversarial attacks against visual perception pipelines in autonomous driving. Both subprocesses in the visual perception pipeline, object detection and multiple object tracking (MOT), are considered. The paper proposes a novel approach in adversarial attacks, the tracking hijacking, which can fool the MOT process using Adversarial Examples (AEs) in object detection. The key idea is to exploit the tracking error to place specific attacks on single frames in MOT, which can lead to a displacement of the detected objects. It is shown that the proposed method can effectively attack the perception pipeline by just fooling 2 to 3 consecutive frames on average.

Advantages
The paper is well written and the topic generally of interest to the ICLR community. The paper also provides a nice overview on the processing pipeline. Technically, it appears relatively straight foward; the main contribution lies perhaps in the interplay of detection and tracking that may render this approach interesting to practitioners.

Comments
The authors state that smaller thresholds R and H are tested since those are more conservative in real-world scenarios. However, success rates for both (R, H) settings are shown in Figure 4. It does not become obvious whether smaller values for R and H values are tested in addition.

The authors state that a previous attack is only effective when AE can reliably fool at least R consecutive frames. Since it is more conservative in current real-world development to use smaller thresholds R and H, the previous attack could also be successful fooling the object detection. But even if the threshold is set to R = 60 and H=6, tracking hijacking is shown to be successful by just fooling 3 frames. This fact, however, is not fully prominent in the text. It would be helpful to describe this in more detail in the effectiveness section.

Minor comments:
Proofreading is necessary (e.g., top of page 6)
Repetitions of sentences:
- "[...] successful AEs on as few as one single frame, and 2 to 3 consecutive frames on average [...]" (p.2, p.2, p.3, p.4)
- "attack success rate (100%) [...] few as 3 frames" (twice on p.8)

Rating: Borderline

Post rebuttal: Thanks for the clarification and the numbers. I am raising the score to weak accept.

**Experience Assessment:**

I have read many papers in this area.

**Review Assessment: Checking Correctness Of Derivations And Theory:**

I assessed the sensibility of the derivations and theory.

**Review Assessment: Checking Correctness Of Experiments:**

I assessed the sensibility of the experiments.

**Review Assessment: Thoroughness In Paper Reading:**

I made a quick assessment of this paper.

---

> ### Author Response · Authors · 2019-11-11
> **Authors' Response**
>
> Thank you for you comments and suggestions!
>
> - Attack evaluation under smaller R and H settings
>
> In general, larger R and H values in the MOT can better correct errors in the detection, but at the same time will also result in longer-lasting attack effects under our tracker hijacking attack. We evaluated two sets of (R,H) shown in figure 4(b), where (5, 2) is considered as the smaller (R,H) settings adopted in real-world self-driving code base. We will make it more prominent in writing that our attack is far more effective than previous attack under recommended settings (60, 6), while even under (5, 2) where the tracker hijacking effects becomes less persistent, it is still better than previous attack that targets solely on detection models.
>
> We indeed have performed experiments on even smaller (R,H), but did not show the results previously since they are unlikely to be used in practice due to the further decreased effectiveness in correcting errors in the detection step. For example, the results for (1, 1), the extreme case of smaller (R, H), are shown below. As shown, the takeaway is similar to that by comparing the results of (5,2) and (60, 6) in figure 4(b): when (R, H) becomes smaller, detection attack requires less consecutive frames to succeed, but our tracker hijacking attack still outperforms it by achieving better success rate with 2 attack frames.
>
>
> Tracker hijacking attack (success rate %)
> ——————————————————————--
> # of frames to fool      1       2       3        4         5         6        7          8
> R=1,H=1,cov=0.1       30    100     100    100     100    100    100     100
> R=1,H=1,cov=1          5      55      100     100     100     100    100    100
> R=1,H=1,cov=10        5      25      100     100     100     100    100    100
>
> Object detection attack (success rate %)
> ——————————————————————--
> # of frames to fool      1       2       3       4       5       6       7       8
> R=1,H=1,cov=0.1         5      5     100    100     100    100     100     100
> R=1,H=1,cov=1            5      5      100     100     100     100     100    100
> R=1,H=1,cov=10          5      5      100     100     100     100     100    100

---

### Official Review · AnonReviewer3 · 2019-10-24
**Official Blind Review #316**

**Rating:** 8

**Review:**

The paper studies adversarial attacks against MOT and proposes a tracker hijacking attack that can successfully fool the MOT with a high success rate even when only adding patches to 3 consecutive frames.

This work is the most realistic attack I have seen so far on attacking the visual perception system of the self-driving car since it not only considers object detection but also the object tracking, which is critical for autonomous driving system. The paper is well-motivated and illustrates the importance of the problem.

According to my knowledge, this is the first paper to study the complete visual perception pipeline in autonomous driving. Previous work along this line focuses on attacking the object detection task, however, this work considers object tracking, which is essential for a safety-critical system as temporarily failure of object detection for one frame usually won’t hurt the decision. Thus, it is an important step towards a more realistic attack on the visual perception pipeline in autonomous driving.

The paper also presents a thorough background of MOT. Their proposed algorithm works much better than other alternatives.

Weakness:
I wish the evaluation can be conducted a bit more broadly. Currently, it is only evaluated 20 clips from one dataset.

Minor points
In Eqn (1), \lambda and \gamma are not defined.
Page 5 last line, “associate” should be “associated”.


**Experience Assessment:**

I have published one or two papers in this area.

**Review Assessment: Checking Correctness Of Derivations And Theory:**

I did not assess the derivations or theory.

**Review Assessment: Checking Correctness Of Experiments:**

I assessed the sensibility of the experiments.

**Review Assessment: Thoroughness In Paper Reading:**

I read the paper at least twice and used my best judgement in assessing the paper.

---

> ### Author Response · Authors · 2019-11-11
> **Authors' Response**
>
> Thank you for your valuable feedback!
>
> - Concerns regarding the scale of our evaluation
>
> We agree that it is better if we can increase the scale of our current evaluation by experimenting on more video clips. One technical challenge we are facing in achieving this is the manual efforts required in annotating the patch area in each video frame. To maximize the realism in our evaluation, we limit the patch for adversarial perturbations to only a small area (size of a license plate) on the back of the vehicle (usually located below the plate), and to simulate the realistic setting, we are not fixing the position across all the video frames. Instead, we manually annotate the coordinates of patch area for each video frame to be always consistent with the vehicle position, which means for a 2s video of 30 fps, we currently need to manually draw 60 bounding boxes. We will clearly acknowledge this limitation in our paper, and identify potential directions in automating this manual process, e.g., leveraging object detections, as future work.

---

### Decision · Program_Chairs · 2019-12-19

**Decision:**

Accept (Poster)

**Comment:**

The authors agree after reading the rebuttal that attacks on MOT are novel.  While the datasets used are small, and the attacks are generated in digital simulation rather than the physical world, this paper still demonstrates an interesting attack on a realistic system.